# Female copulation song is modulated by seminal fluid

Peter Kerwin[1], Jiasheng Yuan[1] & Anne C. von Philipsborn [1✉]

In most animal species, males and females communicate during sexual behavior to negotiate reproductive investments. Pre-copulatory courtship may settle if copulation takes place, but often information exchange and decision-making continue beyond that point. Here, we show that female *Drosophila* sing by wing vibration in copula. This copulation song is distinct from male courtship song and requires neurons expressing the female sex determination factor DoublesexF. Copulation song depends on transfer of seminal fluid components of the male accessory gland. Hearing female copulation song increases the reproductive success of a male when he is challenged by competition, suggesting that auditory cues from the female modulate male ejaculate allocation. Our findings reveal an unexpected fine-tuning of reproductive decisions during a multimodal copulatory dialog. The discovery of a female-specific acoustic behavior sheds new light on *Drosophila* mating, sexual dimorphisms of neuronal circuits and the impact of seminal fluid molecules on nervous system and behavior.

---

[1] Danish Research Institute of Translational Neuroscience (DANDRITE), Nordic-EMBL Partnership for Molecular Medicine, Aarhus University, Hoegh-Guldbergsgade 10, 8000 Aarhus, Denmark. ✉email: avp@mb.au.dk

Sexual behavior provides outstanding models for circuit neuroscience and behavioral genetics, enabling the study of how multimodal communication modulates innate drives and action patterns[1–6]. Male gametes are usually less costly than female gametes, making male pre-copulatory courtship widespread. In the fruit fly Drosophila melanogaster, males sing a close range courtship song by vibrating one extended wing, giving rise to trains of pulses interspersed by a sinusoidal hum. Initiation of song behavior as well as its proper species-specific motor patterning is under the control of sexually dimorphic neuronal circuits expressing the sex determination factors Fruitless (FRU) and Doublesex (DSX), which have been mapped and characterized in great detail[4–9]. The song of a courting male increases the probability of a female to accept him for copulation[4–6].

Transfer of sperm and seminal fluid during copulation decreases the female's propensity to engage in subsequent matings. This effect is mainly mediated by products of the male accessory glands[10–13]. Despite this chemical mate guarding, females mate multiple times when in social groups and in the presence of food[14–17]. The outcome of sperm competition in multiply inseminated females depends on genotypes of all involved parties and female remating rate[11,12,15–17]. It is also impacted by plastic copulation behavior of the male[18,19]. Like in many other species, including birds and mammals[20,21], male flies can strategically allocate ejaculate to optimize investment in mating partners depending on social context and female condition[19,22–25]. Potential responses of Drosophila females to these male tactics are not well understood, but females are predicted to modulate post-mating physiology and behavior in a context-dependent manner[19].

We discovered that Drosophila females produce pulsed wing vibrations in copula, which are acoustically distinct from male pre-copulatory courtship song. Here, we investigate this novel acoustic behavior, its neuronal basis and sensory control, and demonstrate that it has an effect on reproductive competition.

## Results

### Female Drosophila sing a sex specific song during copulation.

In audio recordings of Drosophila melanogaster, we noticed regular sound pulses during copulation, which were distinct from pre-copulatory male song (Fig. 1a). Intermittent female bilateral wing vibration during copulation was reported previously[26]. We confirmed the occurrence of such wing vibrations by high-speed video, and found them to coincide with sound pulses (Supplementary Movie 1). Clipping both wings of females completely abolished copulation sound pulses (n = 67 couples with wing clipped female). We thus conclude that females sing a wing song in copula. Copulation song is seen during the vast majority of copulations of wild type flies (97.5%, n = 200 couples, 11.5 ± 11.5 pulses/min copulation, mean ± s.d.). Copulation song pulses have higher fundamental frequency than male pulse song, multiple cycles and are spaced at longer intervals (Fig. 1b, Supplementary Audio 1). They occur throughout the entire 15–20 min of copulation, with a higher probability at the beginning and very end of copulation (Fig. 1c, Supplementary Fig. 1a). The acoustic properties of song pulses did not change over the course of copulation (Supplementary Fig. 1b). We did not observe copulation song in single females, grouped females or before the start of copulation, but sometimes recorded few song pulses immediately after a couple had disengaged (12% of copulations, n = 50, Supplementary Fig. 1a). Female song pulses come in trains, with 50% of the inter pulse intervals shorter than 200 ms (Fig. 1d). Females of the closest relatives of D. melanogaster, D. simulans, D. mauritiana and D. sechellia, also sing copulation songs (Fig. 1e–g, Supplementary Fig. 1c–e), highlighting that the behavior is

evolutionarily conserved. Copulation song in D. simulans (seen in 88.5% of copulations, n = 26 couples, 0.9 ± 0.9 pulses/min copulation, mean ± s.d.) and D. mauritiana (seen in 28.1% of copulations, n = 64 couples, 1.8 ± 2.6 pulses/min copulation, mean ± s.d.) is less frequent than in D. melanogaster. Median inter pulse intervals, pulse frequency and pulse cycles do not significantly differ between D. melanogaster, D. simulans and D. mauritiana females (Kruskal–Wallis tests with Dunn's multiple comparison, Fig. 1e–g). Copulation song of D. sechellia (seen in 89.1% of copulations, n = 46 couples, 6.7 ± 5.9 pulses/min copulation, mean ± s.d.) has shorter inter pulse intervals, lower fundamental pulse frequency and more pulse cycles than D. melanogaster song (p < 0.0005, Kruskal-Wallis tests with Dunn's multiple comparison, Fig. 1e–g, Supplementary Fig. 1f). The acoustic parameters for D. sechellia copulation song are in accordance with an earlier report of the behavior in this species[27]. In D. simulans and D. sechellia, female copulation song also occurs with higher probability at the beginning of copulation (Supplementary Fig. 1f, g). In all four species, female copulation songs differ significantly from male pre-copulatory songs in inter pulse intervals, pulse carrier frequency and pulse cycles (p < 0.0001, Mann-Whitney tests, Fig. 1e–g). We next wondered if females also sang during interspecific matings. Approximately 7% of D. melanogaster females accepted D. sechellia males in our song recording chambers. In 79% of these copulations (n = 19), we detected female copulation song (4.4 ± 8.3 pulses/min copulation, mean ± s.d.), which resembled more closely copulation song of D. melanogaster intraspecific matings than song of D. sechellia intraspecific matings (Fig. 1f, g). Pulse frequencies and pulse cycles were slightly higher in interspecific matings than in D. melanogaster intraspecific matings (p = 0.0006 and p = 0.003, respectively, Mann-Whitney tests). This might reflect a bias in the subset of D. melanogaster females accepting D. sechellia males or changes in mechanical properties of the couples.

### Neuronal control of female copulation song.

We next asked if female wings are passively shaken in copula or actively vibrated by the female neuromuscular apparatus. Wing vibrations during flight and male song require activity of the motor neurons innervating the flight power muscles (indirect dorsal longitudinal motor neurons, dlm mns)[28]. Silencing of dlm mns with tetanus toxin (TNT) abolished copulation song, indicating that song is actively produced by females (Fig. 2a, b). Silencing of neurons expressing the female specific sex-determination factor DoublesexF (DSXF), which have been implicated before in female reproductive behaviors[29–31], likewise led to loss of copulation song. dsx positive (dsx+) neurons do not comprise dlm mns or other wing muscle motor neurons, but sensory and interneurons[29] (Fig. 2b). They thus might be pre-synaptic to dlm mns and involved in patterning female song or detecting sensory stimuli from the copulating male. Among dsx+ neurons, neurons negative for sex specific fruitless transcripts (fru−) and located in the ventral nerve cord (otd−) were required for song, whereas silencing of dsx+ fru+ neurons did not affect song (Fig. 2a, b). When females were beheaded immediately after the onset of copulation, the majority of them still sang (53%, n = 1 5, 2.6 ± 2.0 pulses/min copulation, mean ± s.d., Supplementary Fig. 2a). These data suggest that ventral nerve cord neurons are sufficient for generating copulation song. Brain neurons might still be important for initiating or modulating singing. Silencing of all brain dsx+ neurons led to very low female receptivity (Supplementary Table 1). We therefore separately tested the role of two prominent dsx+ brain neuronal classes, pC1 and pCd, which have been shown to influence female receptivity[30]. Silencing of neither pC1 nor pCd affected copulation song (Supplementary Fig. 2b).

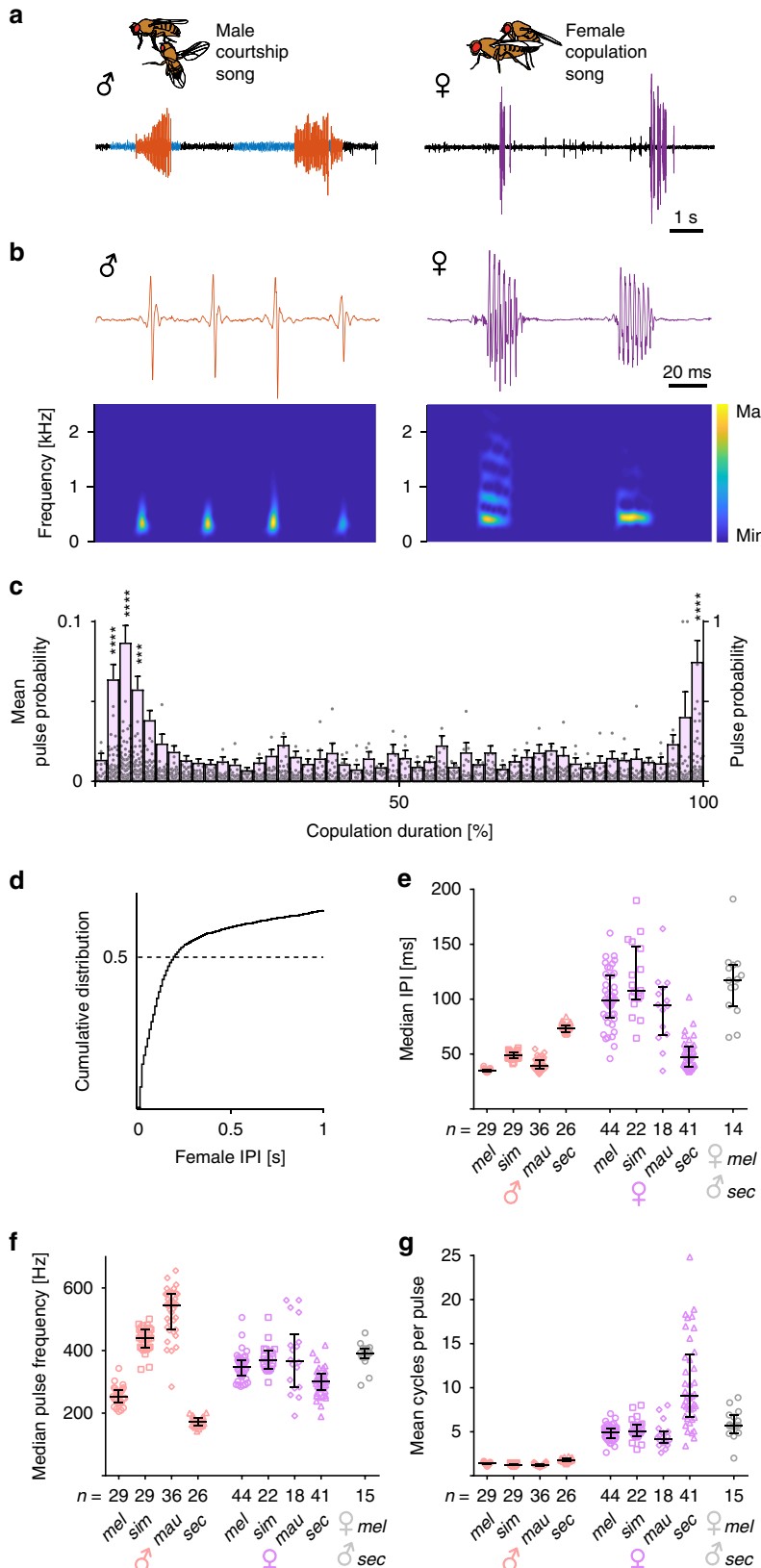

**Female copulation song depends on seminal fluid transfer.** Male song commences when males touch the female abdomen during courtship and taste pheromones[4,6]. We wondered which sensory stimuli could trigger female copulation song and if the mating male had an influence on female singing. Wild type males which were immature (24 h after eclosion), small or depleted of seminal fluid by multiple recent matings elicited less female copulation song than control males on their first mating (Fig. 3a). These male conditions could have affected intensity and quality of male pre-copulatory courtship. In a data set of matings between wild type partners, we did not find the amount of female song correlated with either the amount of pre-copulatory male song or

**Fig. 1 Female *Drosophila* sing during copulation. a** Oscillograms of male pre-copulatory courtship song (pulse song in red, sine song in blue) and female copulation song (magenta), with typical postures during the behavior. **b** Male and female song pulses with spectrograms. **c** Mean probability of female song pulses throughout copulation (magenta bars depicting 2% bins of total copulation duration, left y-axis), and pulse probability of individual flies in 2% bins of total copulation duration (gray data points, right y-axis). $n = 92$ copulations, each data point represents one fly, error bars indicate mean and s.e.m., ****$p < 0.0001$, ***$p = 0.00015$ permutation test (one-sided). **d** Cumulative distribution of female song inter pulse intervals (IPI), with 50% of IPIs shorter than 200 ms. $n = 4543$ IPIs from 44 females. **e–g** Acoustic parameters of male and female pulse song for *D. melanogaster* (mel), *D. simulans* (sim), *D. mauritiana* (mau), *D. sechellia* (sec) and *D. melanogaster* females mating with *D. sechellia* males. Each data point represents one fly, *n* is indicated under the graph, error bars indicate median and interquartile range. Source data are provided as a Source data file.

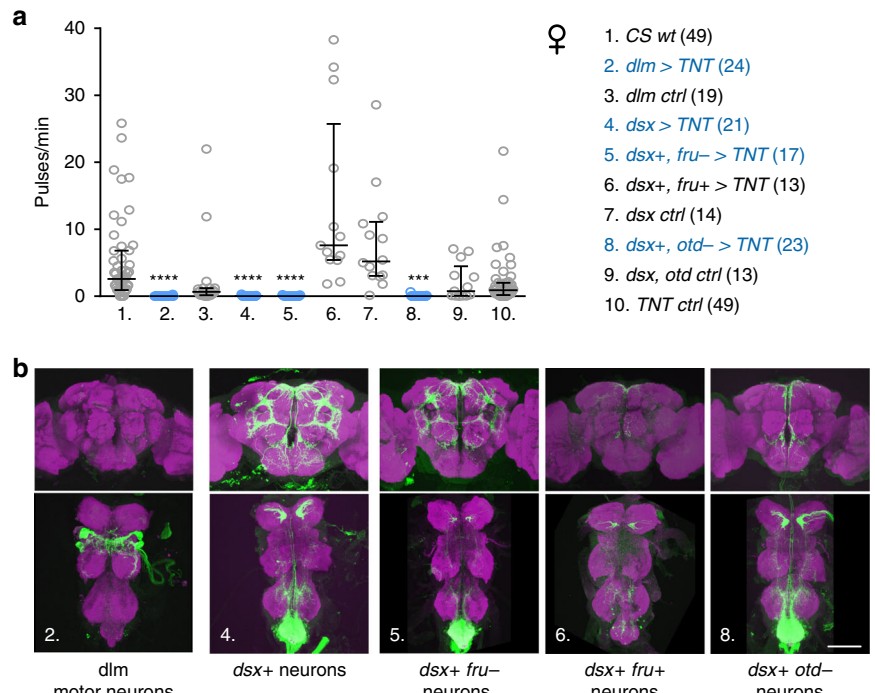

**Fig. 2 Neuronal control of female copulation song. a** Song pulses per minute copulation (with *CS wt* males) for females with different subsets of neurons silenced with tetanus toxin (TNT) and controls. For full genotypes, see Methods. Significant reduction of female song is indicated in cyan. Each data point represents one fly, n is indicated after the genotypes, error bars indicate median and interquartile range, ****$p < 0.0001$, ***$p = 0.0003$, Kruskal–Wallis test with Dunn's multiple comparison (two-sided). **b** Nervous system expression pattern of selected genotypes from **a**, Gfp in neurons in green, neuropil anti-bruchpilot staining in magenta. A micrograph of one representative tissue (out of 5 or more samples per genotype with the same expression pattern) is shown. Scale bar, 100 μm. Source data are provided as a Source data file.

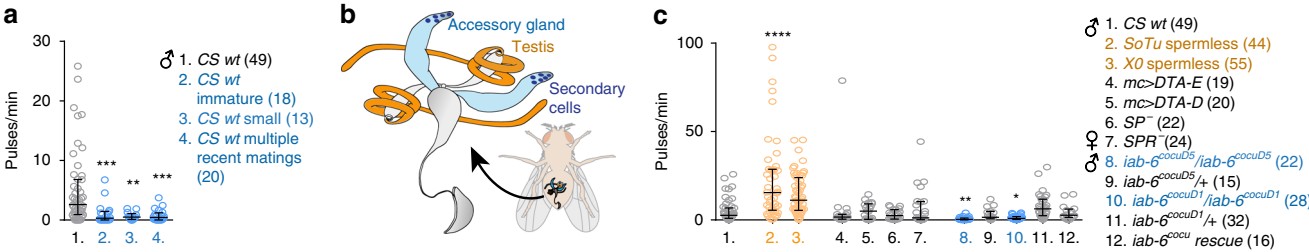

**Fig. 3 Female copulation song depends on seminal fluid transfer. a** Song pulses per minute copulation (of wild type females), mated to wild type males in different conditions. Significant reduction of female song is indicated in cyan. Each data point represents one fly, n is indicated after the conditions, error bars indicate median and interquartile range, ***$p = 0.0007$ (2.), **$p = 0.0017$ (3.), ***$p = 0.0002$ (4.), Kruskal–Wallis test with Dunn's multiple comparison (two-sided). **b** Schematic of male internal reproductive organs, with accessory glands in cyan and testes in orange. **c** Song pulses per minute copulation (of wild type females or females mutant for sex peptide receptor (SPR)), mated to males with sperm production or accessory gland secretions impaired and respective controls. For full genotypes, see Methods. Significant increase of female song is indicated in orange, reduction in cyan. Each data point represents one fly, *n* is indicated after the genotypes, error bars indicate median and interquartile range, ****$p < 0.0001$, **$p = 0.001$, *$p = 0.0385$, Kruskal–Wallis test with Dunn's multiple comparison (two-sided). Source data are provided as a Source data file.

the latency to copulation (Supplementary Fig. 3a). Manipulation of male song amplitude by clipping the distal half of both wings did not have any effect on subsequent female singing. Likewise, females deprived of visual cues (courtship and copulation under dim red light or *white* mutant females), females unable to smell the male pheromone cVA and other odors detected by the odorant receptor family (*orco1* mutant[32,33], lacking the olfactory co-receptor Or83b) or females with their external chemosensory bristles transformed into taste-blind mechanosensory bristles (*poxn70−23* mutant[34]) all sang similar amount of copulation song as did wild type females exposed to unaltered sensory cues from their partners (Supplementary Fig. 3b). From these data, we conclude that it is unlikely that male pre-copulatory courtship has a large impact on female copulation song. Immature, small and multiply mated have impaired ejaculate quality or quantity[15,35,36]. We thus hypothesized that sensory stimuli from ejaculate components evoked or enhanced female song. Previously, reproductive tract neurons expressing the mechanosensory channels Pickpocket 1 (Ppk1)[37] or Piezo[38] have been implicated in the female response to copulation. Neither *ppk1Δ16* mutant females[39] nor *piezoKO* mutant females[40] sang less copulation song than controls (Supplementary Fig. 3c). Silencing female LSAN interneurons, which relay Piezo signals during copulation[38], also had no effect on copulation song (Supplementary Fig. 3d). We conclude that Ppk and Piezo dependent mechanosensation is not required for copulation song. Male fly ejaculate is composed of sperm, produced in the testes, as well as seminal fluid, which is secreted by the accessory gland and other tissues of the reproductive tract[12] (Fig. 3b). Both *XO*[41] and *son of tudor*[42] males, which lack sperm, but not seminal fluid, elicited more female copulation song than control males (Fig. 3c). Transfer of seminal fluid without sperm is therefore sufficient for copulation song. Among seminal fluid molecules, products of the accessory gland influence female physiology and behavior[10–13]. The accessory gland consists of two types of secretory cells, which differ morphologically and biochemically[13,43]. To distinguish between the effect of main and secondary secretory cells of the accessory gland, we tested males with diphtheria toxin ablated main cells (*mc > DTA*)[42] as well as *iab-6cocu* mutant males, which have defective secondary cells[13,44,45]. Disruption of secondary, but not main cell function strongly reduced the amount of female copulation song elicited by a mating male. In line with this, abolishment of transfer or detection of sex peptide (SP), which is produced by main cells and unaffected in *iab-6cocu* mutants[44], did not affect female song (Fig. 3c). We therefore conclude that secondary cell products (SCPs) are necessary for female singing and might be the sensory cue triggering female song.

**Female copulation song influences remating.** The discovery of a female specific copulation song raises the question of its functional relevance. We wondered if it acted as a cue for male mating partners and could influence copulation behavior and ejaculate transfer. Copulation song of wild type females varies in amount. Amount of copulation song was not correlated with copulation duration (Supplementary Fig. 4a). Copulation durations of couples with mute females (wing amputated or dlm mns silenced) or deaf males (aristae amputated) did not differ from controls (Supplementary Fig. 4b). These data suggest that copulation song does not affect copulation duration. To further probe for effects, we compared copulations of wild type males with mute females that were either supplemented by playback of copulation song or left in silence (Fig. 4a). The duration of copulation (Fig. 4b), the number of adult progeny resulting from it (Fig. 4c) as well as the number of sperm in female storage organs (Fig. 4d) did not differ between conditions. Next, we challenged the mating by removing

the first male after the natural termination of copulation and presenting the mated female with the opportunity to mate with subsequent males for the following 6 days (Fig. 4e). To assess remating, we counted the number of females with progeny from subsequent males. For four days, females from couples that had received playback remated significantly less than controls (Fig. 4f). Overall, females produced the same number of total progeny irrespective of whether song was played back or not during the first mating (Fig. 4g). However, the difference in remating latency translated to a significantly increased relative reproductive advantage of the first male in the song playback group compared to silence control (Fig. 4h). To confirm the effect of female song on remating with wild type flies, we performed the playback experiment with wild type females muted by bilateral wing amputation and assessed remating with subsequent wild type males by videotaping fly interactions for the first 20 h following the first mating. With these genotypes, over 80% of females from both playback and silence groups remated within the first two days. At 3 h after the first mating, a lower percentage of females from the playback group had remated than from the silence group (Fig. 4i). Likewise, the latency to the first remating within 20 h was significantly longer in the playback group (Fig. 4j). We conclude that female song playback during the first mating delays subsequent remating of females muted either by wing mn silencing or wing amputation.

Female remating is strongly modulated by seminal fluid components, among these sex peptide as well as other accessory gland proteins (Acps), some of which interact with sex peptide[10–13,46]. We therefore hypothesize that males adjust quantity and/or composition of transferred seminal fluid upon hearing copulation song (Fig. 4k). We used the paired (prd) driver to express Gfp-tagged sex peptide (Gfp-SP) in the male accessory gland[47], and quantified the transfer of fluorescent protein to females. With this method, we did not observe any difference in Gfp-SP levels in the female reproductive tract in song playback vs. silence groups. The size of the mating plug did also not differ (Supplementary Fig. 5). The difference in remating might thus not be due to increased SP transfer during song playback, but depend on other seminal fluid molecules. Alternatively, a significant effect might only be detectable with more sensitive methods for quantifying endogenous SP along with the transferred seminal fluid proteome.

## Discussion

In the present study, we report a novel acoustic behavior during *Drosophila* reproduction, female specific copulation song. It occurs in *D. melanogaster* as well as in its sibling species *D. simulans*, *D. mauritiana* and *D. sechellia*. While the acoustic parameters of male courtship song display marked inter-species differences, female song structure in *D. melanogaster*, *D. simulans* and *D. mauritiana* species is very similar. This is in line with the proposed function of male song as a prezygotic isolating barrier[48]. In contrast, such a function seems unlikely for female copulation song, which occurs after a mating partner has been chosen. Our results revise the notion that only male *Drosophila melanogaster* sing[5,6]. We identify neuronal components of a female song circuit with shared and distinct elements compared to its male counterpart. Silencing of a specific motor output neuronal class required for flight, dlm mns, abolishes female song completely, whereas it only decreases the amplitude of male song[28]. Male song depends critically on *dsx+ fru+* neurons, many of which are male specific or sexually dimorphic[4,7–9]. For female song, we find *dsx+ fru−* neurons of the ventral nerve cord, but not *dsx+ fru+* neurons necessary for song. Future work dissecting thoracic circuits will reveal how sex specific wing motor patterning is

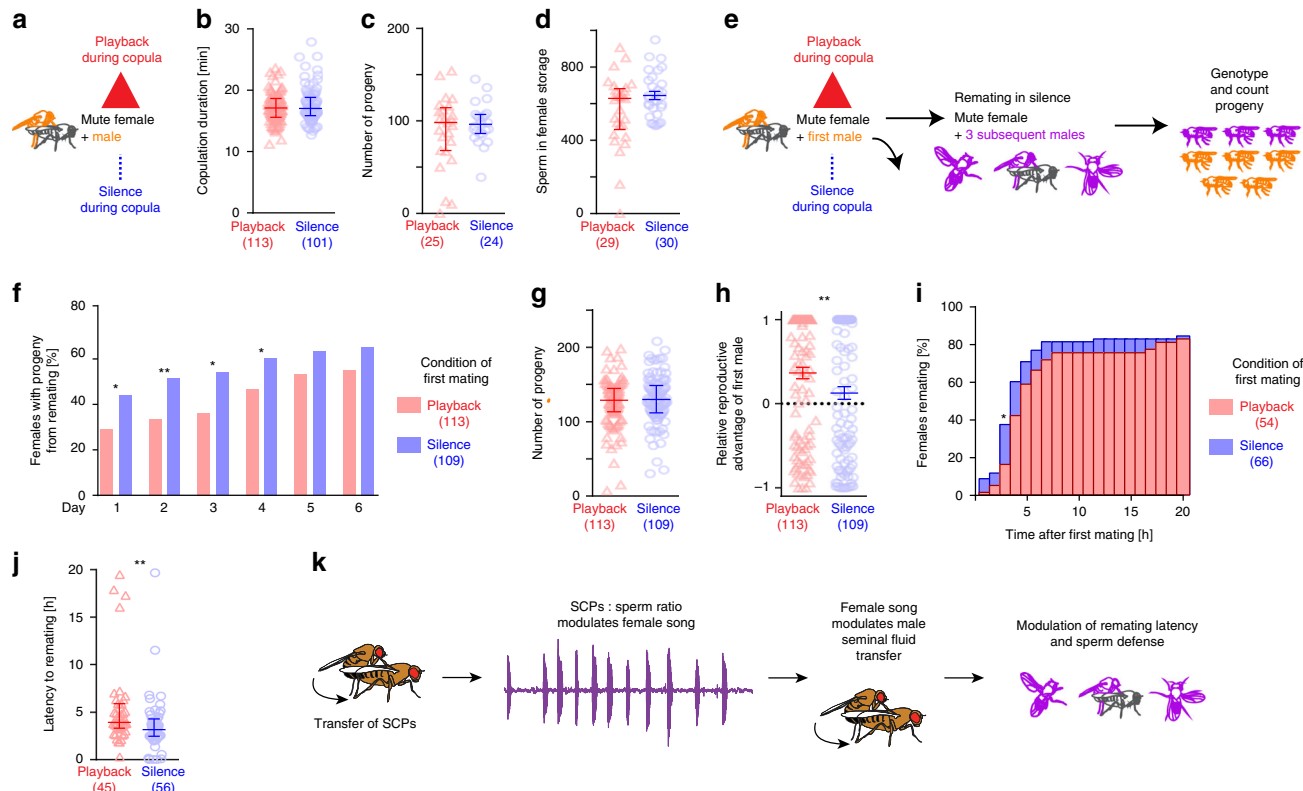

**Fig. 4 Female copulation song decreases remating. a** Experimental design for testing the effect of female song. Data from the playback condition is shown in red, from the control silence condition in blue. **b** Copulation duration of mute females with wild type males, with and without female song playback. **c** Adult progeny after 6 days of egg laying from copulations of mute females with wild type males, with and without female song playback. **d** Number of sperm in female storage organs after copulations with and without female song playback. **e** Experimental design for testing the effect of female song playback under subsequent reproductive competition. The first male and it's progeny is displayed in orange, the subsequent males and their progeny in magenta. **f** Female remating after copulations with and without song playback. *p = 0.0258 (day 1), **p = 0.0097 (day 2), *p = 0.0102 (day 3), *p = 0.0446 (day 4). Fisher's exact test (two-sided). **g** Adult progeny after 6 days of egg laying, from copulations of mute females with wild type males, with and without female song playback, with the female subsequently housed with 3 males carrying genetic markers *Sb/ Tm3, Ser*. **h** Relative reproductive advantage of the first male ((progeny first male−progeny subsequent males)/all progeny)) in the experiment outlined in **e** after 6 days of egg laying. n is indicated after the conditions, error bars indicate standard error of the mean, **p = 0.0099, permutation testing (one-sided). **i** Percentage of wild type females muted by wing amputation remating within 20 h with wild type males after copulations with and without song playback. *p = 0.014, Fisher's exact test (two-sided). **j** Latency to the first remating after copulations with and without song playback. **p = 0.0035, Mann-Whitney test (two-sided). **k** Schematics of proposed copulatory dialog. In **b**, **c**, **d**, **g** and **j** each data point represents one couple, *n* is indicated after the conditions, error bars indicate median and interquartile range, no significant difference in **b**, **c**, **d** and **g**, Mann-Whitney test (two-sided). Source data are provided as a Source data file.

generated and how neuronal dimorphisms explain the different acoustic parameters of male and female song.

We find the composition of male ejaculate, critically affects female singing behavior, whereas the quality of male pre-copulatory courtship (visual, olfactory and gustatory input) is likely to have little impact. Absence of sperm in the presence of seminal fluid increases singing. Males generally depleted of ejaculate or specifically lacking SCPs in their seminal fluid barely elicit any female song. More song in the absence of sperm could be due to a potential increase in seminal fluid or a greater accessibility of SCPs, some of which are normally bound to sperm[11,12,46]. Alternatively, sperm might suppress female singing. Sperm is transferred in a discrete, ~1 min long bout around 7–8 min after start of copulation. In contrast, seminal fluid transfer starts immediately after the initiation of copulation and is thought to continue until disengagement[47,49–51]. This transfer pattern might explain the higher probability of female song at the beginning and end of copulation.

Since females mutant for two mechanosensory channels expressed in the female reproductive tract (Ppk and Piezo) still sing in copula, it is unlikely that transfer of ejaculate elicits female

singing via mechanical stimulation. We hypothesize that SCPs provide a chemical cue for the female, analogous to female pheromones triggering male courtship. It will be interesting to unravel if a single SCP or a more complex mixture is needed for female song initiation, by which receptor(s) and sensory neurons the transfer is detected and how the signal is relayed to motor patterning circuits.

Our experiments demonstrate that female copulation song influences female remating and reproductive success when females have the possibility to remate. How could hearing female song exert such an effect? Female remating probability is impacted by the composition of male ejaculate[10–13,44]. The latter is not fixed, but can be modulated by the male to adjust the amount of seminal fluid and sperm transferred to the presence of rivals and female condition (so-called strategic ejaculate alloca-tion)[19,22–25]. Since seminal fluid is depleted after several matings, strategic allocation has been predicted to be adaptive for males[19,22]. Based on these previous findings, we propose that female copulation song directly affects male seminal fluid allo-cation and by this decreases female remating. Since we see an effect of song playback, we assume that males detect female song

with their auditory system. However, this does not rule out that males can also detect female song by mechanosensation. Female wing vibrations during singing might also dissipate pheromones and thereby influence olfaction.

We further propose that a feedback loop might coordinate female singing and fine-tune ejaculate transfer. At the beginning of copulation, male SCPs trigger female song, cueing the male that his partner is responsive to seminal fluid components. During the subsequent course of copulation, female song influences further ejaculate transfer. Here, the function of female song could be to entice allocation of costly components from the male. Alternatively, female song might help to proportion overall ejaculate composition to match individual physiological needs (Fig. 4k). Females might not be able to predict male ejaculate composition by assessing male pre-copulatory courtship. There is no evidence that females can prematurely terminate copulations. Copulation song might thus be a way for females to give feedback to and influence males with whom they have chosen to copulate, modulating allocation behavior to their benefit.

Our first investigation of female copulation song has not yet unraveled its evolutionary significance, and we can only speculate about potential roles in sexual conflict and sexual selection. We found evidence that female song can delay remating. So far, we do not know if this is the only or most important function of female singing or might be only a secondary effect of changed ejaculate composition. Delayed remating could be adaptive for females under certain conditions, when it leads to efficient use of the sperm from the first male before it is replaced by the ejaculate of a subsequent mate[16]. This might be in the interest of females, since it allows for mixed paternity and genetic diversity of their offspring[17].

In our working model, the SCPs stimulating female singing are not necessarily identical with the seminal fluid components that are differentially transferred. In the future, comprehensive screening of the numerous SCPs which are altered in expression levels in iab-6cocu mutants[13], as well as analysis of seminal fluid composition in song playback vs. silence copulations by ELISA[52] or quantitative proteomics[53–55] are needed to test these hypotheses. Identifying the factors which are differentially transferred in response to female song is crucial for building hypotheses about the adaptive value of female song.

In general, it can be in the interest of females to influence ejaculate allocation, receipt of seminal fluid components, and, ultimately, paternity of their offspring by active signaling[19,56–59]. We propose copulation song as a new mechanism by which male reproductive competition, most likely via male allocation behavior, can be influenced by females.

## Methods

**Fly strains**. Flies were raised on cornmeal, oatmeal, yeast, sucrose food under a 12 h:12 h light:dark cycle at 25 °C and 60–70% humidity. *D. sechellia* cultures were supplemented with Formula 4–24 Instant *Drosophila* Medium, Blue (Carolina Biological Supply Company) dissolved in noni juice (nu3 GmbH). The following fly strains were used: *D. melanogaster* wildtype Canton S (*CS wt*), *D. simulans* wild type (gifts from B. Dickson), *D. mauritiana* wild type (Kyoto DGRC, 900020), *D. sechellia* wild type (DSSC 14021–0248.07, gift from T. Auer and R. Benton), *dlm-SG* split GAL4 driver[28], UAS-TNT (Bloomington Bl28837), UAS-CD8-Gfp (gift from B. Dickson), dsx-GAL4 (Bl66674), LexAop-Gal80 (gift from B. Dickson), fru-LexA (Bl66698), UAS-FRT-stop-FRT-TNT (Bl67690), fruFLP (Bl66870), Otd-nlsFLPo (gift from K. Asahina and D. Anderson), tubP-FRT-stop-FRT-GAL80 (Bl39213), tubP-FRT-GAL80-FRT (Bl38881), tsh-GAL80 (gift from B. Dickson), GMR71G01.LexA.attp40 (Bl54733), GMR41A01.LexA.attp40 (Bl54787), LexAop-FLP (Bl55819), tud1, bw, sp (Bl1786), c(1)RM/ C(1;Y)6, y, w, f/ 0 (Bl9460), mc > DTA-E and mc > DTA-D (gifts from M. Wolfner), Df(1)Exel6234 (Bl7708), SP[0] (Bl77892), Df(3L)Δ130 (gift from J.C. Billeter), iab-6cocuD5, iab-6cocuD1 and iab-6cocu rescue (gifts from F. Karch), orco[1] (Bl23129), poxn[70–23] (gift from U. Heberlein), ppk1[Δ16] (gift from Y.N. Jan), piezo[KO] (Bl58770), UAS-Kir2.1 and LSAN-1 split GAL4 driver (ss03919, gifts from L. Shao and U. Heberlein), +; +;

Sb/Tm3, Ser (*CS wt* background, gift from B. Dickson), protamineB-Gfp (Bl58404), prd-GAL4 (Bl1947), UAS-Gfp-SPc (gift from M. Wolfner and T. Aigaki).

Full genotypes of transgenic experimental flies are for Fig. 2: +; UAS-TNT, UAS-CD8-Gfp/ GMR23H06.p65ADZp.attp40; GMR30A07.ZpGAL4DBD.attp2 (dlm > TNT), +; GMR23H06.p65ADZp.attp40; GMR30A07.ZpGAL4DBD.attp2 (dlm ctrl), +; UAS-TNT, UAS-CD8-Gfp; dsx-GAL4 (dsx > TNT), +; UAS-TNT, UAS-CD8-Gfp/ LexAop-Gal80; fru-LexA/ dsx-GAL4 (dsx+, fru− > TNT), +; UAS-FRT-stop-FRT-TNT; dsx-GAL4/ fruFLP (dsx+, fru+ > TNT), +; +; dsx-GAL4 (dsx ctrl), w; UAS-TNT, UAS-CD8-Gfp/ Otd-nlsFLPo; tubP-FRT-stop-FRT-GAL80/ dsx-GAL4 (dsx+, otd− > TNT), w; UAS-CD8-Gfp/ Otd-nlsFLPo; tubP-FRT-stop-FRT-GAL80/ dsx-GAL4 (dsx ctrl), +; UAS-TNT, UAS-CD8-Gfp; + (TNT ctrl). Supplementary Fig. 2: Genotypes for targeting pC1 and pCd were as described previously:[30] w; GMR71G01.LexA.attp40/ UAS-FRT-stop-FRT-TNT; dsx-GAL4/ LexAop-FLP (pC1 > TNT), w; GMR71G01.LexA.attp40; dsx-GAL4 (pC1 ctrl), w; GMR41A01.LexA.attp40/ UAS-FRT-stop-FRT-TNT; dsx-GAL4/ LexAop-FLP (pCd > TNT), w; GMR41A01. LexA.attp40; dsx-GAL4 (pCd ctrl), w; UAS-FRT-stop-FRT-TNT; LexAop-FLP (TNT, Flp ctrl). Figure 3: SoTu spermless males were the male progeny of homozygous tud1, bw, sp females crossed to CS wt males, XO spermless males were the male progeny of c(1)RM/ C(1;Y)6, y, w, f/0 males crossed to CS wt females. SP- males were as described previously:[10] SP[0]/ Df(3L)Δ130, SPR- females were homozygous for Df(1)Exel6234. Supplementary Fig. 3: piezo[KO] flies were in CS wt background. LSAN neurons were targeted as described previously:[38] w; LSAN1, UAS-Kir2.1 (LSAN1 > Kir2.1), w; LSAN1 (LSAN1 ctrl), w; +; UAS-Kir2.1 (Kir2.1 ctrl). Figure 4: mute females were w-; UAS-TNT, UAS-CD8-Gfp/ GMR23H06.p65ADZp.attp40; GMR30A07.ZpGAL4DBD. attp2 (dlm > TNT), males for quantification of sperm were +; protamineB-Gfp/ +; + (CS wt background), subsequent rival males carrying genetic markers had the genotype+; +; Sb/ Tm3, Ser. Supplementary Fig. 4: +; UAS-TNT, UAS-CD8-Gfp/ GMR23H06.p65ADZp.attp40; GMR30A07.ZpGAL4DBD.attp2 (dlm > TNT), +; GMR23H06.p65ADZp.attp40; GMR30A07.ZpGAL4DBD.attp2 (dlm ctrl), +; UAS-TNT, UAS-CD8-Gfp; + (TNT ctrl), Gfp-SP fluorescence in the female bursa was quantified from mute dlm > TNT females mated to males of the genotype w; UAS-Gfp-SPc, prd-GAL4.

**Sound recording, playback and progeny count experiments**. All flies used in behavioral experiments were collected after eclosion and aged for 4–7 days. Virgin females were aged in groups of 10–20 flies in standard culture vials. Males were aged in isolation in flat bottom 1.5 ml 96 well blocks filled with 0.5 ml food per well and covered with a PCR foil with air holes. For song recording, We used a multi-channel array of electret condenser microphones (CMP-5247TF-K, CUI Inc), amplified with a custom-made circuit board and digitized with a multifunction data acquisition device (NI USB-6259 MASS Term, National Instruments)[8,28]. For determining when flies were copulating, simultaneous video recording was performed. For high speed recording of female wing vibrations during copulation, we used an Optronis CR3000x2, monochrome camera, equipped with a Sigma 105 mm f2.8 macro lens. For recording copulation song from headless females (Supplementary Fig. 2a), couples were allowed to copulate and female were quickly beheaded with fine spring scissors. For this operation, the copulating pair was very gently collected with an aspirator (with a cut 1 ml pipette tip at the end) shortly after the onset of genital coupling. The aspirator was placed under a stereomicroscope and the scissors were held to the opening of the pipette tip. The aspiration caused the female to move, allowing for beheading at the moment her head emerged from the tip opening. The copulating pair was then transferred to a recording chamber.

To obtain small males used in Fig. 3a, flies were grown under crowded conditions at larval stage and specimen that were shorter than 2 mm (anterior posterior body axis) were selected at eclosion. Males were depleted of seminal fluid by pairing them for 5–6 h with 8–10 CS wild type virgin females (4–6 d old). Depletion was confirmed by testing if the subsequent recorded copulation was fertile. Only copulations with no offspring were further evaluated. For female song playback experiments, we chose a part of a recording of a representative wild type female with an amount of song pulses in the highest quartile of a distribution of 200 females (205 s long, 35.7 pulses/min, looped). Flies were allowed to copulate in double-netted chambers (1 cm diameter, 4 mm height) and female song was played back 10 min during copulation. The sound intensity of female pulses at the location of the chambers was 60–90 dB, with most pulses at 75–78 dB, as measured with the noise meter Decibel X (SkyPaw). Playback sound intensity was calibrated beforehand to behaviorally relevant levels as follows: a part of the same recording containing male courtship song (before copulation, 3 s long) was played back at the respective amplitude and found to rescue decreased copulation success of wingless males (68% of CS wt pairs copulating within 30 min during playback vs. 4% of pairs copulating in silence, n = 28 for each condition p < 0.0001, Fisher's exact test). After playback, mated females were collected into individual food vials and kept for 6 days either alone or in the presence of 2–3+; +; Sb/ Tm3, Ser males. After 10 days of the playback experiment, progeny was counted and genotyped for 6 days. For playback experiments with wing amputated CS wt females, female wings were removed with fine spring scissors one day after eclosion and females were allowed to recover for 3 days. After playback, mated females were collected into individual food vials with 2–3 CS wt males and time lapse videotaped (at 80 s intervals) for 20 h.

**Immunohistochemistry and microscopy**. Nervous systems were dissected in ice cold PBS, fixed for 30 min in 4% paraformaldehyde and stained for 2 days at 4 °C with nc82/anti-bruchpilot antibody (mouse, Developmental Studies Hybridoma Bank, Cat# nc82; RRID:AB_2314866, 1:50) and anti-GFP antibody (rabbit, Torrey Pines Biolabs, Cat# TP401 071519; RRID:AB_10013661, 1:6000), followed by a 2 day incubation at 4 °C with secondary antibodies Goat anti-Mouse IgG Alexa Fluor 647 and Goat anti-Rabbit IgG Alexa Fluor 488 (Thermo Fisher Scientific, Cat# A-21236; RRID: AB_2535805 and Cat# A-11034; RRID: AB_2576217). Tissues were mounted vectashield (Vector Labs, Cat# H-1000). [8,28]. Confocal image stacks were collected on a Zeiss LSM 780 with a Zeiss Plan-NeoFluar 25×/0.8 multi immersion objective. For counting protamineB-Gfp labeled sperm, females were frozen at −20 °C 55 min after the end of copulation. Spermathecae and seminal receptacles were dissected in ice cold PBS, and slightly squished or uncoiled for mounting, respectively. For imaging Gfp-SP and mating plug (PEB-me[51]) auto-fluorescence, females were frozen at −20 °C within 40 min after the end of copulation. Bursae were dissected and mounted in ice cold PBS. Female reproductive tissues were imaged on a Zeiss Axio Imager M2 with a Plan Apochromate 20×/0.8 and a Plan Apochromate 10×/0.45 objective. Sperm number was determined in ImageJ with the Cell Counter plugin. Fluorescence intensity was measured with Zen Blue software (Carl Zeiss Microscopy).

**Song analysis, data evaluation and statistical analysis**. Male song was detected by a MATLAB script (*FlySongSegmenter*, by B. Arthur and S. Stern, https://github.com/FlyCourtship/FlySongSegmenter), oscillograms and tentative annotations were then visualized and corrected manually by visual and auditory inspection[8,28]. Female song pulses were annotated manually by a combination of visual inspection and listening to pitch and analyzed by custom written MATLAB scripts. Cyclicity of pulses was assessed by the minimum of positive and negative pulse peaks, counting all peaks with at least 2/3 the amplitude of the largest peak within a pulse. For IPI analysis, all IPIs between 15–200 ms were evaluated. Statistical analyses were performed in MATLAB (MathWorks) and Prism6 (GraphPad Software). Details about statistical tests are provided in the figure legends. If not indicated otherwise, each fly was treated as independent sample and the statistical tests were two-sided. For evaluating the distribution of song pulses during the length of copulation (Fig. 1c, Supplementary Fig. 1g, h), wild type copulations were normalized to time and binned into 2% intervals of total copulation duration. For each bin, the average probability of a song pulse was calculated across all flies. For statistical testing, the 50 bins were permuted randomly for each fly 100,000 times. p-Values for the bins at the beginning and end of copulation report the probability for a bin mean at any time point exceeding the observed experimental value. For evaluating the significance of the relative reproductive advantage reported in Fig. 4h, individual females and their genotyped progeny counts were randomly assigned to playback and silence condition (100,000 permutations). p values for each day report the probability for the difference of mean relative reproductive advantages of the first male exceeding the observed experimental value.

**Reporting summary**. Further information on research design is available in the Nature Research Reporting Summary linked to this article.

## Data availability

The source data underlying Figs. 1c–g, 2a, 3a, c, 4b–d, f–j and Supplementary Figs 1b, g, h, 2b, 3a–d, 4a, b and 5 are provided as a Source data file. Additional raw data comprising song recording audio files, video files and confocal micrograph stacks are available upon request from the corresponding author.

## Code availability

MATLAB scripts used for analysis are available upon request from the corresponding author.

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

## Acknowledgements
We thank Duda Kvitsiani, Poul-Henning Jensen, Jean-Christophe Billeter and Young-Joon Kim for helpful discussions, Tatiana Adamiec, Anna Prudnikova and Markéta Kaderávková for technical assistance, Barry Dickson, Mariana Wolfner, Toshiro Aigaki, Jean-Christophe Billeter, François Karch, Robert Maeda, Richard Benton, Thomas Auer, Lisha Shao, Ulrike Heberlein, Yuh Nung Jan as well as Bloomington and Kyoto DGRC Drosophila stock centers for fly stocks. This study was supported by Lundbeckfonden grant DANDRITE-R248-2016-2518.

## Author contributions
A.C.v.P. and P.K. designed the study; A.C.v.P., J.Y. and P.K. performed experiments and analyzed the data. A.C.v.P. wrote the manuscript with input of P.K.

## Competing interests
The authors declare no competing interests.
