## [Peer Review File · Nature Communications]

Reviewers' Comments:

Reviewer #1:

Remarks to the Author:

Review of the paper by Kerwin, Yuan and Philipsborn

The current study aims to unravel a novel aspect of *Drosophila melanogaster* female sexual behaviour modulating reproduction. The authors have analyzed and shown that during copulation, the female produces a vibration with her wings. They have compared the female "song" in two closely related species (*D.simulans*, *D.mauritiana*). The vibration specifically occurs at the very beginning and at very end of the copulation period (Figure 1). Using classical genetic mutations and transgenes, the authors have inactivated some sex specific thoracic and abdominal ganglia neurons governing female wing pulse (Figure 2). They also measured the intensity of the female song with males either subjected to a variable number of copulation events, or depleted for some factors—sperm or accessory gland products—transferred to the female during copulation (Figure 3). In the last shown series of experiments (Figure 4), they tested the effect of a female synthetic song played back during copulation in the presence of mute females. They also tested male competition under the same mute/playback condition with several labelled males successively paired with the same mute female.

In general, I found the paper very clear and easy to follow. The finding is quite interesting and I was very excited when I first read the study. However, I feel that the data shown do not fully support the main conclusion made by the authors. There are also some points that need to be clarified, including some of those leading to the main conclusions of the paper.

Major points:

The key finding which the authors used to propose a model regarding the function of female song on reproduction is shown on Figure 4f. However, the examination of the two series of data (silence/playback) shown on that figure can lead to a different interpretation compared to that stated by the authors. Is it possible to imagine that mutant males carrying two non-behaviourally neutral mutations (*Stubble* and *Serrate*) carried on a *TM3* balancer chromosome (a large inversion of most of the genetic material on that chromosome) are more sensitive to the "song" than the initial male, and therefore are disturbed to transmit slightly less sperm than under the "silence" condition? Moreover, one gets puzzled when analyzing more closely the data of this Fig.4f. Indeed, both distributions largely overlap and only diverge for a small fraction of males. Although the picture of the figure is not so clear on my computer, I can guess that only 10-12% of the males (among about 110 individuals) diverged between the two groups. This means that the variation induced by the female synthetic song would only differentially affect a small fraction of the tested males.

Moreover, the behavioural interpretation of the effect induced by the female song could be different from that proposed here. Based on my above comment (on Fig.4f), how can the authors exclude the possibility that the effect of the wing beat is more mechanical than acoustic? Given that the female vibrates wings at the two extreme time points (beginning/end) of copulation, could one hypothesize that the initial series of vibration allows the male fly to stabilize and get centered—between the wing beats—over the female back, while the final series of vibrations would provide a strong signal to indicate the end, or even to kick out the male out of copula? In this case, the female would still play an active role in copulation duration, but the function of her vibrations would be different from that proposed here. Such hypothesis can be tested using mutant males either deaf or mechanosensory-insensitive to female wing beat. Moreover, but to a lesser extent, female wing beat could also serve to propagate the less volatile female cuticular pheromones and therefore change the physiological activity of the male in copula.

During copulation the two partners mutually exchange multiple signals and some of them can have multiple functions (such as male wing vibration). In the case investigated here, the female song could be induced by male activity when copulating, and in the same time affect male transfer of seminal factors(s) during copulation. In relation to such bidirectional interaction between the sexes, the authors should clarify the fact that while spermless males (Fig.3c/#2-3) induced higher wing beat frequency in females, the song playback increased the number of sperms transferred by the "first" male (Fig 4f)?

Minor comments:

Page 2, L37-38: "female ...passive players". I refute this, given that several references highlighting the critical role played by the female before and during copulation can be found in the literature.

Page 3, L52 and L61-63: the inter-species comparison shows very different values for the vibration parameter (which are not "very similar" as written P6, L123). This raises a question on the evolutionary conservation and function of this "song" in related *Drosophila* species. In the other two species, was this increased amount of vibration also observed at the two extreme copulation time points? What happens when a *D.simulans* female copulates with a *D.melanogaster* male? This could be tested.

Reviewer #2:

Remarks to the Author:

In this study, Kerwin et al. has described a novel female courtship behavior in fruitfly: female singing via wing vibration during copulation. They found that production of female copulation song depends on seminal fluid protein but not sperm; and proposed that female courtship song affects reproduction by influencing males' ejaculate allocation. They also looked into some circuit mechanisms. Overall, this work is very exciting in revealing an overlooked role of female in acoustic communication during copulation and will be of broad interests to both neuroscientists and evolutionary biologists.

Major comments:

(1) A key claim of this study is that female copulation song affects male sperm defense, based on two results: first, copulation song depends on seminal fluid proteins, which is essential for sperm defense; second, in a setting where female is forced to disengage the first male and then given the option to mate a second male, female copulation song playback can increase relative offspring number of the first male. However, there are multiple gaps in this claim. First, female copulation song depends on seminal fluids doesn't mean seminal fluid is the only, or even is, the factor that has been modulated by female's song during copulation. Second, a more direct assay is required to measure the consequences of female song in transferring sperm and Acps. Sperm can be measured by using GFP-expressed sperm. Acps can be measured either biochemically (See Wolfner's work), or by GFP-tagged sex peptide, or by the size of mating plug (under UV light) as the minimal. These are all available tools or previously described experiments. Third, instead of counting offspring number, female's remating frequency is a much more direct measurement for the sperm defense mechanism that the authors have acclaimed.

(2) To describe this behavior more thoroughly, how female singing is elicited is an important topic. The authors state in discussion that without excluding the possibility of a mechanosensory mechanism, "our data supports the hypothesis that SCPs provide a chemical cue for the female, analogous to female pheromones triggering male courtship". However, there is no evidence to support

it at all. In fact, since female song happens in seconds after copulation starts, mechanosensory mechanism is a completely valid and even tempting hypothesis. To explore the mechanism further, the authors can use Piezo or ppk mutant to test the relevant hypothesis (ref to Shao et al. 2019 Neuron).

(3) The authors have used immature male, small male, and mated males to demonstrate that male's quality affect female song production. However, It is not clear if females make active decision in singing or just respond passively to males' ejaculate, which results in a feedback loop between the male and female. Do females evaluate males' quality pre-copulatively during courtship, which influences females' decision in singing? This question is critical for understanding the evolutionary significance of the behavior. Perturbation experiments that test different variables in courtship behavior (e.g., cutting off male's wing to abolish singing) are needed to address it.

(4) Finally, I'm afraid that the authors didn't make very far in terms of the underlying neuronal mechanisms. In my opinion, this is OK for the moment being and I'm sure more circuit mechanisms will be revealed as they dive in (e.g, what neurons are sensing the stimuli, how this information is relayed, and how the motor pattern is generated, how this pathway is potentially modulated by female' mating status, and etc). My major concern is that I don't find their conclusion (dsx+fru- VNC neurons trigger female copulation song) very convincing. It is made based on that inhibiting dsx+fru+ or dsx+otd- neurons don't affect female song production. Unfortunately, the intersection between dsx and otd failed to rule out the contribution of brain neurons completely (Fig. 2b). Moreover, the authors claim that "this copulation song is ... generated by neurons expressing female sex determination factor Dsx^F". This is not factually wrong, but the word "generate" can be potentially inaccurate. In fact, In VNC, the expression of Dsx is restricted to hundreds of Abg neurons and a few interneurons, and there is no detected expression in the wing neuropil at all (see Zhou et al. Neuron 2014 and others). Last, the authors summarized that "these data suggest that the neuronal circuits for male and female song are particularly overlapping, but distinct". I found that this is a big statement with little information.

Minor comments:

(1) The authors have observed (1) song pulses after male and female disengages, and (2) increase of putative copulation song approaching the end of copulation. I'm wondering if (2) actually signals the attempt to end copulation. A comparison of detailed sound features between the beginning of copulation and the end of copulation (or a time serial analysis of song features) will help to confirm that they are actually the same signal.

(2) The authors only analyzed three D. species, in which D. simulans and D. mauritiana are sibling species that represents a single ancestral lineage in relative to D. melanogaster. Inspection of more species (especially ones representing independent lineages) are required to make a proper augment that female songs are less variable than male song (line 64).

(3) It will be interesting to see if sensory stimuli from male SCPs is sufficient to maintain female courtship song by cutting off female's head after copulation starts, if possible.

(4) "Playback volume was calibrated beforehand to behaviorally relevant levels with male courtship song by rescuing decreased copulation success of wingless males (line 263)." It is not clear why playback volume was calibrated in ref to male's song versus female's song and how exactly the calibration is done.

Reviewer #3:

Remarks to the Author:

This paper uncovers an interesting phenomenon that Drosophila females produce song-like acoustic signals by bilateral wing vibration during copulation. The authors claim that these are female song, and distinct from the male courtship song that is generated by fru/dsx neurons in males, the female

song is generated by neurons expressing DsxF. They also proposed that these female songs are triggered by the transfer of seminal fluid components, and males who hear these female songs can increase the reproductive success rate when challenged by competition, which may be due to increased sperm defense. Overall, this is an interesting study that uncovers potentially hidden roles of female song during mating, but I have major concerns regarding the quality and conclusion of this study.

Major concerns:

1. The first major concern is whether the acoustic signals recorded from bilateral wing flicks during copulation could be called SONG, and thus are comparative to male courtship song, for the purpose of communication during mating/courtship. Distinct from male courtship song that has almost fixed song parameters (e.g. sing song frequency, IPI) for the purpose of effective communication, acoustic signals produced by females during mating are very noisy: IPIs ranging from <50ms to >150ms, and mean cycles per pulse ranging from 2 to 7, in individual females. Could such different acoustic signals produced by individual females be called song for the effective purpose of communication during mating? Could these acoustic signals just randomly generated by wing flicks (e.g. for grooming purpose, and triggered by mechanosensory inputs during mating)?

2. The authors proposed that "female song" were triggered by receipt of seminal fluid components, but to my knowledge, transfer of seminal fluids happens a couple of minutes after the start of copulation, but the authors show that "female song" produced in higher probability in the beginning (e.g. within the first minute, fig. 1c) and the end of mating process. If seminal fluids are not transferred in the first minute during mating, it is wrong to claim that it's the seminal fluid components that trigger "female song". Not mentioning that the authors did not test any known seminal fluid protein in this process.

3. The function of such "female song" during mating is not convincing. It is better to test wild-type flies to make such statement. Instead the authors used *dIm>TNT* females (the mute females as they called), and the ONLY tested genotype for such a crucial statement. Furthermore, there is no difference regarding to the copulation duration and number of offsprings by these *dIm>TNT* females whether there is song playback. The only difference is from fig. 4f but is not convincing (look at the distribution of 100+ samples), although they are statistically different.

Minor concern:

The authors show that *dsx* neurons control this "female song" and show that expressing TNT in all *dsx* neurons or *dsx+ otd-* neurons in the VNC eliminates song production. These data are fine but how about the *dsx*-expressing neurons in the brain? Since there are published reagents for labeling all *dsx*-expressing neurons in the brain (Koganezawa et al., 2016, The Neural Circuitry that Functions as a Switch for Courtship versus Aggression in *Drosophila* Males), pCd and pC1 neurons (Zhou et al., 2014, Central Brain Neurons Expressing doublesex Regulate Female Receptivity in *Drosophila*), it is reasonable to test the role of these neurons for female song production.

Response to reviewers

We would like to thank the reviewers for very constructive and helpful comments and thoughtful suggestions. With their input, we believe we significantly improved the manuscript.

Summary of revisions:

Text:

We revised the manuscript text to describe and discuss the added data. Some paragraphs were reformulated to make them clearer and correct misperceptions. 22 additional references are added. We also would like to change the title of the manuscript to **“Female copulation song is modulated by seminal fluid”**. We believe the new title is more appropriate, since most of the added data further supports this statement, whereas our mechanistic understanding of copulation sound function in reproductive competition, as pointed out by the reviewers, is still limited.

Figures:

We added new data to main Figures 1, 3 and 4 as well as to Supplementary Figure 1. There are four new Supplementary figures.

Additional experiments and data analysis:

Figure 1 and Supplementary Figure 1:

Copulation song from an additional species, *D. sechellia*.

Copulation song from interspecific matings between *D. melanogaster* and *D. sechellia*.

Analysis of potential changes of song pulse parameters over the course of copulation.

Analysis of song pulse distributions over the course of copulation in *D. simulans* and *D. sechellia*.

Supplementary Figure 2:

Copulation song from headless females.

Copulation song upon silencing of *dsx*+ brain neurons pC1 and pCd.

Figure 3 and Supplementary Figure 3:

Copulation song upon manipulation of sex peptide signaling.

Analysis of correlation between female singing and male courtship.

Copulation song after manipulation of male courtship cues.

Copulation song after manipulation of female mechanosensation and LSAN interneurons.

Figure 4 and Supplementary Figure 4:

Analysis of correlation between copulation singing and copulation duration.

Copulation duration after manipulation of female singing.

Sperm count in females after song playback vs silence.

Analysis of remating after song playback vs silence.

Supplementary Figure 5:

Transfer of Gfp-tagged sex peptide and size of mating plug after song playback vs silence.

We also provide a source data file.

Detailed response to reviewers:

Reviewers' comments:

Reviewer #1 (Remarks to the Author):

Review of the paper by Kerwin, Yuan and Philipsborn

The current study aims to unravel a novel aspect of *Drosophila melanogaster* female sexual behaviour modulating reproduction. The authors have analyzed and shown that during copulation, the female produces a vibration with her wings. They have compared the female “song” in two closely related species (*D.simulans*, *D.mauritiana*). The vibration specifically occurs at the very beginning and at very end of the copulation period (Figure 1). Using classical genetic mutations and transgenes, the authors have inactivated some sex specific thoracic and abdominal ganglia neurons governing female wing pulse (Figure 2). They also measured the intensity of the female song with males either subjected to a variable number of copulation events, or depleted for some factors—sperm or accessory gland products—transferred to the female during copulation (Figure 3). In the last shown series of experiments (Figure 4), they tested the effect of a female synthetic song played back during copulation in the presence of mute females. They also tested male competition under the same mute/playback condition with several labelled males successively paired with the same mute female.

In general, I found the paper very clear and easy to follow. The finding is quite interesting and I was very excited when I first red the study. However, I feel that the data shown do not fully support the main conclusion made by the authors. There are also some points that need to be clarified, including some of those leading to the main conclusions of the paper.

Major points:

The key finding which the authors used to propose a model regarding the function of female song on reproduction is shown on Figure 4f. However, the examination of the two series of data (silence/playback) shown on that figure can lead to an different interpretation compared to that stated by the authors. Is it possible to imagine that mutant males carrying two non-behaviourally neutral mutations (Stubble and Serrate) carried on a TM3 balancer chromosome (a large inversion of most of the genetic material on that chromosome) are more sensitive to the "song" than the initial male, and therefore are disturbed to transmit slightly less sperm than under the “silence” condition? Moreover, one gets puzzled when analyzing more closely the data of this Fig.4f. Indeed, both distributions largely overlap and only diverge for a small fraction of males. Although the picture of the figure is not so clear on my computer, I can guess that only 10-12% of the males (among about 110 individuals) diverged between the two groups. This means that the variation induced by the female synthetic song would only differentially affect a small fraction of the tested males.

We changed Figure 4 as well as the paragraph referring to the figure. Our new data shows that playback vs silence does not lead to changes in the number of sperm stored in female seminal receptacle and spermathecae shortly after copulation (Fig. 4d). We also show that females from copulation receiving playback remate less with subsequent (Stubble Serrate) males than silence controls (Fig. 4f). This

differential remating explains the difference in reproductive advantage (former Fig. 4f, now shown in Fig. 4h) and the fact that only a subset of flies are responsible for the effect (that is, for non-remating females the first male has a reproductive advantage of 1, and a higher number of these is present in the playback group. Females remating only on day 2 or later have more cumulative progeny from the first male after 6 days as well). We would like to thank the reviewer for a very discerning eye in this respect. The raw data is now also provided in the source data file.

We agree that Stubble Serrate Tm3 Balancer chromosome males might behave differently from wild type males. We did not detect any obvious impairment in their copulation behavior. In our experimental design, we aim at testing the effect of song on the first male, which always was wild type. If there is a specific effect of Stubble Serrate Tm3 Balancer chromosome males, this effect should be the same in playback and silence condition, since the Stubble Serrate Tm3 Balancer chromosome males mate as subsequent males in both conditions in the absence of female song. We added the label “Remating in silence” to the experimental design schematics (Fig. 4e) to present our assay in a clearer way.

Moreover, the behavioural interpretation of the effect induced by the female song could be different from that proposed here. Based on my above comment (on Fig.4f), how can the authors exclude the possibility that the effect of the wing beat is more mechanical than acoustic? Given that the female vibrates wings at the two extreme time points (beginning/end) of copulation, could one hypothesize that the initial series of vibration allows the male fly to stabilize and get centered—between the wing beats—over the female back, while the final series of vibrations would provide a strong signal to indicate the end, or even to kick out the male out of copula? In this case, the female would still play an active role in copulation duration, but the function of her vibrations would be different from that proposed here. Such hypothesis can be tested using mutant males either deaf or mechanosensory-insensitive to female wing beat. Moreover, but to a lesser extent, female wing beat could also serve to propagate the less volatile female cuticular pheromones and therefore change the physiological activity of the male in copula.

We can definitely not exclude the possibility that the effect of the wing beat has mechanical components, but could not find a way of playing these back in a controlled way. The “mute” females we used have the indirect flight musculature paralyzed, and therefore cannot vibrate their thorax and in this way provide mechanosensory input to the male (as a female with clipped wings might do). To test the copulation stabilization hypothesis, we looked more carefully into a potential connection between copulation duration and female singing. We do not find a correlation between amount of female song and copulation duration (Supplementary Fig. 4a), and different types of “mute copulations” (wingless females, deaf males, females with wing muscles paralyzed) are not shorter in duration or aborted at early stages.

The possibility that female song pulses at the beginning and end of copula might be functionally different has been also pointed out by reviewer 2. We investigated this idea and do not see any difference in acoustic parameters for song pulses at different times during copulation (Supplementary Fig. 1b). From new graphs displaying all raw song pulse data (Supplementary Fig. 1b), it is also more apparent that many female song trains occur in the middle of copula without leading to termination of copulation.

We also added a short description of the temporal dynamics of seminal fluid vs sperm transfer to the discussion, as well as a speculation how these could lead to lower probabilities of song in the middle of copula (line 201-204)

We cannot exclude that female wing beats propagate cuticular pheromones. Since we see an effect of playback, we propose that males can detect song by hearing. We added the mention of potential other detection channels.

During copulation the two partners mutually exchange multiple signals and some of them can have multiple functions (such as male wing vibration). In the case investigated here, the female song could be induced by male activity when copulating, and in the same time affect male transfer of seminal factors(s) during copulation. In relation to such bidirectional interaction between the sexes, the authors should clarify the fact that while spermless males (Fig.3c/#2-3) induced higher wing beat frequency in females, the song playback increased the number of sperms transferred by the “first” male (Fig 4f)?

The newly added sperm count data supports the notion that female song does not have an effect on sperm transfer (Fig. 4d). As pointed out in the discussion, we do not know if spermless males elicit more song because sperm (and some song inhibiting agent connected with it) is absent or because they might transfer more or differently composed seminal fluid.

Minor comments:

Page 2, L37-38: “female ...passive players”. I refute this, given that several references highlighting the critical role played by the female before and during copulation can be found in the literature.

We changed the sentence and added a reference to a review from 2019, which strongly supports the active role of females.

Page 3, L52 and L61-63: the inter-species comparison shows very different values for the vibration parameter (which are not "very similar" as written P6, L123). This raises a question on the evolutionary conservation and function of this “song” in related *Drosophila* species. In the other two species, was the increased amount of vibration also observed at the two extreme copulation time points? What happens when a *D.simulans* female copulates with a *D.melanogaster* male? This could be tested.

We analyzed the distribution of song for *D. simulans* and *D. sechellia* and also see a higher probability of singing at the beginning of copulation in these species (Supplementary Fig. 4g, h). Since most *D. mauritiana* females sing little song (raw data shown in Supplementary Fig. 4d) and our dataset was small due to low copulation rates, we did not analyze distributions for this species.

We thank the reviewer for the excellent suggestion to investigate interspecies matings! Although we were able to obtain copulations between *D. simulans* and *D. melanogaster* when we left large groups of flies for several days on food, we did not succeed in observing copulations in our small song recording chambers. However, we could record copulations between *D. sechellia* males and *D. melanogaster* females. Since we found that *D. sechellia* females sing a copulation song that is different from *D.melanogaster* females, we considered this interspecies combination even more interesting. The interspecies song data is now shown in Fig. 1 e-g.

Reviewer #2 (Remarks to the Author):

In this study, Kerwin et al. has described a novel female courtship behavior in fruitfly: female singing via wing vibration during copulation. They found that production of female copulation song depends on seminal fluid protein but not sperm; and proposed that female courtship song affects reproduction by influencing males' ejaculate allocation. They also looked into some circuit mechanisms. Overall, this work is very exciting in revealing an overlooked role of female in acoustic communication during copulation and

will be of broad interests to both neuroscientists and evolutionary biologists.

Major comments:

(1) A key claim of this study is that female copulation song affects male sperm defense, based on two results: first, copulation song depends on seminal fluid proteins, which is essential for sperm defense; second, in a setting where female is forced to disengage the first male and then given the option to mate a second male, female copulation song playback can increase relative offspring number of the first male. However, there are multiple gaps in this claim. First, female copulation song depends on seminal fluids doesn't mean seminal fluid is the only, or even is, the factor that has been modulated by female's song during copulation.

We apologize for our unclear description of the remating assay. We did not forcibly terminate the first copulation, as our previous wording could suggest. We changed the sentence, adding "after the natural termination of copulation".

The finding that song depends on seminal fluid certainly does not prove or even suggest that song in turn modulates seminal fluid transfer. We regret to have created this impression. We hope that the new sperm count data (Fig. 4d) and changes to the text in results and discussion will make the reasoning which lead to our current working model clearer.

Second, a more direct assay is required to measure the consequences of female song in transferring sperm and Acps. Sperm can be measured by using GFP-expressed sperm. Acps can be measured either biochemically (See Wolfner's work), or by GFP-tagged sex peptide, or by the size of mating plug (under UV light) as the minimal. These are all available tools or previously described experiments. Third, instead of counting offspring number, female's remating frequency is a much more direct measurement for the sperm defense mechanism that the authors have acclaimed.

These are very valid requests and suggestions.

Sperm count data is now in Fig. 4d.

We evaluated transfer of Gfp-tagged sex peptide and the size of the mating plug by fluorescence microscopy (Supplementary Fig. 5). We did not detect any differences. As briefly mentioned in the text, this could be due to caveats in the methodology (expression of Gfp-SP with the Gal4 UAS system). In the future, we aim at measuring Acps in a less biased way (mass spectrometry, ELISA). Unfortunately, it was beyond the scope of the current manuscript to set up these assays (we are not a mass spectrometry lab and could not obtain quickly a comprehensive set of antibodies against Acps).

We now show remating frequency (estimated from number of females with progeny from second male) over six days (Fig. 4f). See also reply to first major point of reviewer 1.

(2) To describe this behavior more thoroughly, how female singing is elicited is an important topic. The authors state in discussion that without excluding the possibility of a mechanosensory mechanism, "our data supports the hypothesis that SCPs provide a chemical cue for the female, analogous to female pheromones triggering male courtship". However, there is no evidence to support it at all. In fact, since female song happens in seconds after copulation starts, mechanosensory mechanism is a completely valid and even tempting hypothesis. To explore the mechanism further, the authors can use Piezo or ppk mutant to test the relevant hypothesis (ref to Shao et al. 2019 Neuron).

We agree that a mechanosensory mechanism for copulation song triggering is an attractive hypothesis, especially after publication of Shao et al. 2019 (published after our first submission). We have now tested both Piezo and Ppk mutants (Supplementary Fig. 3c), as well as the LSAN interneurons relaying Piezo dependent signals from the female reproductive tract (described in Shao et al. 2019). Since we do not see any effect, we are still inclined to our original hypothesis of chemosensation.

(3) The authors have used immature male, small male, and mated males to demonstrate that male's quality affect female song production. However, It is not clear if females make active decision in singing or just respond passively to males' ejaculate, which results in a feedback loop between the male and female. Do females evaluate males' quality pre-copulatively during courtship, which influences females' decision in singing? This question is critical for understanding the evolutionary significance of the behavior. Perturbation experiments that test different variables in courtship behavior (e.g., cutting off male's wing to abolish singing) are needed to address it.

We investigated potential connections between male pre-copulatory courtship and female singing. We looked at correlations within a wild type data set (Supplementary Fig. 3a) and, as suggested, manipulated sensory modalities relevant for courtship (Supplementary Fig. 3b). We do not find evidence that pre-copulatory courtship modulates female copulation singing, and therefore think that female singing is indeed a direct response to male ejaculate composition. The latter might not be detectable in advance by assessing male pre-copulatory courtship. A short discussion of this is added to the text.

(4) Finally, I'm afraid that the authors didn't make very far in terms of the underlying neuronal mechanisms. In my opinion, this is OK for the moment being and I'm sure more circuit mechanisms will be revealed as they dive in (e.g, what neurons are sensing the stimuli, how this information is relayed, and how the motor pattern is generated, how this pathway is potentially modulated by female' mating status, and etc). My major concern is that I don't find their conclusion (dsx+fru- VNC neurons trigger female copulation song) very convincing. It is made based on that inhibiting dsx+fru+ or dsx+otd- neurons don't affect female song production. Unfortunately, the intersection between dsx and otd failed to rule out the contribution of brain neurons completely (Fig. 2b). Moreover, the authors claim that "this copulation song is ... generated by neurons expressing female sex determination factor Dsx^F". This is not factually wrong, but the word "generate" can be potentially inaccurate. In fact, In VNC, the expression of Dsx is restricted to hundreds of Abg neurons and a few interneurons, and there is no detected expression in the wing neuropil at all (see Zhou et al. Neuron 2014 and others). Last, the authors summarized that "these data suggest that the neuronal circuits for male and female song are particularly overlapping, but distinct". I found that this is a big statement with little information.

We reformulated the paragraph and tested the role of *dsx+* brain neurons (Supplementary Fig. 2b). See also response to the last minor concern of reviewer 3.

Minor comments:

(1) The authors have observed (1) song pulses after male and female disengages, and (2) increase of putative copulation song approaching the end of copulation. I'm wondering if (2) actually signals the attempt to end copulation. A comparison of detailed sound features between the beginning of copulation and the end of copulation (or a time serial analysis of song features) will help to confirm that they are actually the same signal.

This was a very good suggestion. The analysis of song parameters over the course of copulation is now shown in Supplementary Fig. 1b. See also response to second major point of reviewer 1.

(2) The authors only analyzed three *D.* species, in which *D. simulans* and *D. mauritiana* are sibling species that represents a single ancestral lineage in relative to *D. melanogaster*. Inspection of more species (especially ones representing independent lineages) are required to make a proper argument that female songs are less variable than male song (line 64).

We investigated copulation song in *D. sechellia*. We are aware that *D. sechellia* does not represent an independent lineage! Unfortunately, our attempts to record flies from the *virilis* group have not yet been successful (we might have to design bigger recording chambers). Interestingly, *D. sechellia* copulation song is different from the one of its sibling species (Fig. 1e-g). We reformulated the paragraph in the light of this new finding.

(3) It will be interesting to see if sensory stimuli from male SCPs is sufficient to maintain female courtship song by cutting off female's head after copulation starts, if possible.

We very much liked this elegant idea and indeed managed to perform the suggested operation (Supplementary Fig. 2a). The finding that behead female can still sing also supports the notion that brain *dsx+* neurons are not strictly required for song, but might initiate or modulate singing (see reply to major comment above).

(4) "Playback volume was calibrated beforehand to behaviorally relevant levels with male courtship song by rescuing decreased copulation success of wingless males (line 263)." It is not clear why playback volume was calibrated in ref to male's song versus female's song and how exactly the calibration is done.

We specified our procedure in more detail and added measurement of playback intensity in the methods (section: Sound recording, playback and progeny count experiments).

Reviewer #3 (Remarks to the Author):

This paper uncovers an interesting phenomenon that *Drosophila* females produce song-like acoustic signals by bilateral wing vibration during copulation. The authors claim that these are female song, and distinct from the male courtship song that is generated by *fru/dsx* neurons in males, the female song is generated by neurons expressing *DsxF*. They also proposed that these female songs are triggered by the transfer of seminal fluid components, and males who hear these female songs can increase the reproductive success rate when challenged by competition, which may be due to increased sperm defense. Overall, this is an interesting study that uncovers potentially hidden roles of female song during mating, but I have major concerns regarding the quality and conclusion of this study.

Major concerns:

1. The first major concern is whether the acoustic signals recorded from bilateral wing flicks during copulation could be called SONG, and thus are comparative to male courtship song, for the purpose of communication during mating/courtship. Distinct from male courtship song that has almost fixed song parameters (e.g. sing song frequency, IPI) for the purpose of effective communication, acoustic signals produced by females during mating are very noisy: IPIs ranging from <50ms to >150ms, and mean cycles

per pulse ranging from 2 to 7, in individual females. Could such different acoustic signals produced by individual females be called song for the effective purpose of communication during mating? Could these acoustic signals just randomly generated by wing flicks (e.g. for grooming purpose, and triggered by mechnosensory inputs during mating)?

Although IPI and cycle per pulse ranges of female song are broader than for male songs, female songs have a distinct frequency range, which clearly sets them apart from wing flicks and grooming sounds detected during various other behaviors. If female sound pulses were random wing flicks for grooming, it would be unlikely that these were specifically affected by secondary cell secretions. Moreover, we do not see changes in female singing in mechanosensory *ppk* or *piezo* mutants (Supplementary Fig. 3c, also see reply to major comment (2) to reviewer 2).

When investigating an additional species, *D. sechellia*, we realized that female wing vibrations during copula have already been briefly described for this species (Cobb et al. 1989, now added to the references), and were called “copulation song” before. By calling the observed sound pulses “song”, we do not automatically want to claim that they are effective signals for communication. However, we would like to maintain our notion that copulation song can act as a cue, most likely for the male, since we find a significant effect of song playback (Fig. 4f, h).

2. The authors proposed that “female song” were triggered by receipt of seminal fluid components, but to my knowledge, transfer of seminal fluids happens a couple of minutes after the start of copulation, but the authors show that “female song” produced in higher probability in the beginning (e.g. within the first minute, fig. 1c) and the end of mating process. If seminal fluids are not transferred in the first minute during mating, it is wrong to claim that it’s the seminal fluid components that trigger “female song”. Not mentioning that the authors did not test any known seminal fluid protein in this process.

We added more detailed description of the current view about timing of seminal fluid vs. sperm transfer with several references (line 201-204). In the literature, it is documented that seminal fluid transfer starts at the onset of copulation, before the transfer of sperm. This is in line with our hypothesis that secondary cell products elicit singing.

Hundreds of secondary cell proteins are changed in expression in the secondary cell defective *iab-6^{ccu}* mutants used in our study (Sitnik et al. 2016, now added to references). We added a brief discussion of this and pointed out that screening these candidates will test our current hypotheses. One of the most prominent and best-studied seminal fluid protein, sex peptide, is produced by main cells and unaffected in *iab-6^{ccu}* mutants. We therefore used it as a negative control and added song data from copulations of males mutant for sex peptide as well as females mutant for sex peptide receptor (Fig. 3c).

3. The function of such “female song” during mating is not convincing. It is better to test wild-type flies to make such statement. Instead the authors used *dIm>TNT* females (the mute females as they called), and the ONLY tested genotype for such a crucial statement. Furthermore, there is no difference regarding to the copulation duration and number of offsprings by these *dIm>TNT* females whether there is song playback. The only difference is from fig. 4f but is not convincing (look at the distribution of 100+ samples), although they are statistically different.

We decided to use *dIm > TNT* females in combination with playback/silence treatments in order to have full control over the amount of song the copulating wild type male hears, while at the same time keeping the

genotypes of the females the same in the treatment vs control group. This seemed important, since fertility, receptivity and remating rates can vary a lot between different genotypes. We agree that it is not ideal to use non-wild type females in our assay. However, we aimed at testing responses of the wild type male.

As to concern with data in former Fig. 4f (now Fig. 4h), we agree that the effect is moderate. As reviewer 1 pointed out in his first major comment, the effect is due to a subset of tested pairs. We reanalyzed our data to show that differential female remating rates over 4 days lead to the observed relative reproductive advantage of first males that received playback. The data is available in the source data file.

We fully agree that the functional relevance of copulation song needs further investigation. Other functions might still be discovered in the future. In light of this, we also would like to change the title of the manuscript (see above).

Minor concern:

The authors show that *dsx* neurons control this “female song” and show that expressing TNT in all *dsx* neurons or *dsx+ otd-* neurons in the VNC eliminates song production. These data are fine but how about the *dsx*-expressing neurons in the brain? Since there are published reagents for labeling all *dsx*-expressing neurons in the brain (Koganezawa et al., 2016, The Neural Circuitry that Functions as a Switch for Courtship versus Aggression in *Drosophila* Males), pCd and pC1 neurons (Zhou et al., 2014, Central Brain Neurons Expressing doublesex Regulate Female Receptivity in *Drosophila*), it is reasonable to test the role of these neurons for female song production.

We thank the reviewer for this helpful suggestion- similar points were raised by reviewer 2 (see above, major comment(4)). Silencing of all brain *dsx+* neurons (*dsx+ otd+*) led to very low female receptivity. We therefore tested pCd and pC1 separately with the genotypes validated by Zhou et al. 2014 (Supplementary Fig. 2b) and also addressed the role of brain neurons by recording beheaded females (Supplementary Fig. 2b).

Reviewers' Comments:

Reviewer #1:

Remarks to the Author:

All my requests have been dealt with, so to my point of view this paper is acceptable for publication in Nature Communication. I am also admiring of the quantity of novel experiments the authors could carry out since the first version of the paper was submitted.

However, there are some minor points which need to be answered:

- I am curious to know how the authors could behead female during copulation without disturbing this very subtle behavior? They explained in the M&M that they used curved scissors, but still I am admiring of their skills! Could they provide a little bit more explanation?

- Line 104: "data not shown": why?

Lines 220-221: The sentence is unclear and needs to be rewritten.

Reviewer #2:

Remarks to the Author:

I appreciate authors' significant efforts in revising this paper. I think the authors have well addressed most of my questions and concerns, and I believe that the current manuscript is much improved. I only have a few comments:

(1) According to the new data, it appears that females are just passively responding to SCPs, which leaves the evolutionary significance of this behavior very mysterious. For the male's side, it's as if males were using females' response to test their own quality and thus bias his allocation in SCPs to influence re-mating. Shouldn't males try to maximize his reproductive success anyhow? For the female's side, it's even harder to explain. Singing can be energetically costly, what is female benefit from it if there is no space for female choice over mating? The authors' explanation is that it might be a way for females to give feedback to avoid getting stuck. If so, we should expect that when females are muted, the copulation duration takes much longer. I don't think this experiment was directly performed in this paper, but according to the number in 4a, the copulation duration appears quite normal (~18 minutes) when females are muted. Overall, it feels as if some important variables are missing here, or the test that can be applied in the lab may not always be sensitive enough. I think the authors should discuss about the evolutionary significance of this behavior more thoroughly, or directly acknowledge that as something that hasn't been yet explained within the scope of this paper.

(2) Abstract, line 12.

"This copulation song is distinct from the well-studied male courtship song and generated by neurons expressing the female sex determination factor DoublesexF."

I think the more proper statement is that: DsxF neurons are necessary for xxx.

I would be happier to accept the word of "generate" if a small subset of neurons has been shown to be both sufficient and necessary for this behavior. This comment applies for similar statement elsewhere in the paper.

Reviewer #3:

Remarks to the Author:

This revision of the manuscript has made substantial changes, while in some way improves the manuscript, there are some other issues being raised regarding to both the novelty and rigidity of the findings.

1. As stated by the author, Cobb et al., 1989 has identified female copulation song in *Drosophila sechellia* and *Drosophila oreana*, roughly in 250hz and 200hz respectively. This manuscript identified a similar copulation song in *Drosophila melanogaster* and other *Drosophila* species. The identification of *Drosophila* female copulation song is thus not novel.

2. The authors decide to change the title, and not to emphasize the role of this female copulation song in male reproductive competition, probably due to the lack of sufficient evidence. I still feel it is not adequate to make a crucial conclusion based on only one genotype (and it is not wild-type), even without a control genotype. The current title that "female copulation song is modulated by seminal fluid" is more appropriate, but the original novelty by identification of female courtship song and its function in sexual behavior is much compromised.

3. The authors show that silencing all *dsx* neurons eliminated female copulation song, but silencing all *dsx* neurons or only pC1 or pCd neurons in the brain did not affect female copulation song. This raised the possibility that female copulation song is controlled in part by *dsx*-expressing motor neurons in the ventral nerve cord. The authors stated that *dsx* neurons does not comprise dlm mns or other motor neurons (page 96), but without any reference. In fact, *dsx* neurons include motor neurons controlling male courtship song (Shirangi et al., 2013, 2016), it is likely that female copulation song is generated by *dsx*-positive motor neurons, just like in males.

Reply to Reviewers

We are glad we could satisfy many of the previous concerns of the reviewers.

Regarding the remaining/new points, we have changed the manuscript text (indicated in yellow), and added more details to the methods and discussion.

We also added the small supplementary table 1 in the supplementary pdf, which documents female receptivity after silencing of all dsx+ neurons in the brain (3 different genotypes tested).

Please find a point to point reply below.

Reviewers' comments:

Reviewer #1 (Remarks to the Author):

All my requests have been dealt with, so to my point of view this paper is acceptable for publication in Nature Communication. I am also admiring of the quantity of novel experiments the authors could carry out since the first version of the paper was submitted.

However, there are some minor points which need to be answered:

- I am curious to know how the authors could behead female during copulation without disturbing this very subtle behavior? They explained in the M&M that they used curved scissors, but still I am admiring of their skills! Could they provide a little bit more explanation?

We extended the method section to describe the beheading in more detail.

- Line 104: "data not shown": why?

We now give the copulation rates for 3 different genotypes we tested in a supplementary table.

Lines 220-221: The sentence is unclear and needs to be rewritten.

We reformulated the indicated sentence.

Reviewer #2 (Remarks to the Author):

I appreciate authors' significant efforts in revising this paper. I think the authors have well addressed most of my questions and concerns, and I believe that the current manuscript is much improved. I only have a few comments:

(1) According to the new data, it appears that females are just passively responding to SCPs, which leaves the evolutionary significance of this behavior very mysterious. For the male's side, it's as if males were using females' response to test their own quality and thus bias his allocation in SCPs to influence re-mating. Shouldn't males try to maximize his reproductive success anyhow? For the female's side, it's even harder to explain. Singing can be energetically costly, what is female benefit from it if there is no space for female choice over mating? The authors' explanation is that it might be a way for females to give feedback to avoid

getting stuck. If so, we should expect that when females are muted, the copulation duration takes much longer. I don't think this experiment was directly performed in this paper, but according to the number in 4a, the copulation duration appears quite normal (~18 minutes) when females are muted. Overall, it feels as if some important variables are missing here, or the test that can be applied in the lab may not always be sensitive enough. I think the authors should discuss about the evolutionary significance of this behavior more thoroughly, or directly acknowledge that as something that hasn't been yet explained within the scope of this paper.

We agree that there are still aspects of female song that are rather enigmatic. We appended the discussion at several points to add some speculations and/or explanations.

Regarding the question why the male would engage in "strategic ejaculate allocation", as compared to always just give maximum: in the literature, it has been pointed out that seminal fluid is limited and that males might profit from prudent allocation (see line 217-218).

What could be potential benefits for the female? We changed our wording: from "males with whom they are literally stuck with" to "males with whom they have chosen to copulate, modulating allocation behavior to their benefit". We did not want to imply that females wanted to avoid to get stuck in copulation. Rather, we meant that after the decision to accept a male for copulation, females cannot terminate this copulation and have to accept the ejaculate a particular male is able or willing to transfer. They might therefore use copulation song to influence allocation behavior, in order to get an ejaculate that is "good for them". We added some speculation that a certain delay of remating might be adaptive for females, since it allows for more equal usage of sperm from multiple males.

Overall, we agree with the reviewer that there are many aspects of song function that we simply do not understand yet and can only speculate about. We take his/her good advice to state this more clearly in the discussion.

(2) Abstract, line 12.

"This copulation song is distinct from the well-studied male courtship song and generated by neurons expressing the female sex determination factor DoublesexF."

I think the more proper statement is that: DsxF neurons are necessary for xxx.

I would be happier to accept the word of "generate" if a small subset of neurons has been shown to be both sufficient and necessary for this behavior. This comment applies for similar statement elsewhere in the paper.

We change the statement, to "copulation song ...requires neurons expressing DsxF"

Reviewer #3 (Remarks to the Author):

This revision of the manuscript has made substantial changes, while in some way improves the manuscript, there are some other issues being raised regarding to both the novelty and rigidity of the findings.

1. As stated by the author, Cobb et al., 1989 has identified female copulation song in *Drosophila sechellia* and *Drosophila oreana*, roughly in 250hz and 200hz respectively. This manuscript identified a similar

copulation song in *Drosophila melanogaster* and other *Drosophila* species. The identification of *Drosophila* female copulation song is thus not novel.

In Cobb et al. 1989, female copulation song in *D. sechellia* is briefly mentioned, but not further explored. Copulation song has never been described for *D. melanogaster*, *D. simulans* and *D. mauritiana*. We provide the first detailed analysis of the behavior in *D. melanogaster*, the model species used for the vast majority of studies in behavioral and circuit neuroscience. We would like to maintain that there is novelty to this. Further, substantial novelty of the manuscript comes from our findings that (1) female song requires dlm mn and dsx+ neurons, (2) female song requires seminal fluid transfer and function secondary cells of the male accessory gland and (3) female song playback changes postmating behavior.

2. The authors decide to change the title, and not to emphasize the role of this female copulation song in male reproductive competition, probably due to the lack of sufficient evidence. I still feel it is not adequate to make a crucial conclusion based on only one genotype (and it is not wild-type), even without a control genotype. The current title that “female copulation song is modulated by seminal fluid” is more appropriate, but the original novelty by identification of female courtship song and its function in sexual behavior is much compromised.

As pointed out before, we are aware that future studies are needed to fully understand the function of copulation song in sexual behavior. In our first response to the reviewers, we have explained our choice of genotypes, treatment controls and the experimental design underlying the data shown in Figure 4. Our data gives clear evidence that song playback affects remating and paternity in the assay we used. We do not believe that changing the title compromises this evidence.

3. The authors show that silencing all dsx neurons eliminated female copulation song, but silencing all dsx neurons or only pC1 or pCd neurons in the brain did not affect female copulation song. This raised the possibility that female copulation song is controlled in part by dsx-expressing motor neurons in the ventral nerve cord. The authors stated that dsx neurons does not comprise dlm mns or other motor neurons (page 96), but without any reference. In fact, dsx neurons include motor neurons controlling male courtship song (Shirangi et al., 2013, 2016), it is likely that female copulation song is generated by dsx-positive motor neurons, just like in males.

We changed the text from “do not comprise dlm mns or other motor neurons” to “do not comprise dlm mns or other **wing muscle** motor neurons” and now refer more directly to the reference documenting in detail expression pattern of doublesex in males and females (Rideout et al. 2010). In Rideout et al. 2010, it is documented that there are no dsx-positive motor neurons.

We also refer to Figure 2b: As it can be seen there (first nervous system to the left, genotype 2.), dlm motor neurons have big cell bodies and prominent arborisations in the mesothoracic ganglion. These are absent in the stained nervous system displayed next to it (Genotype 4., dsx+ neurons). All other wing muscle motor neurons have their cell bodies in the meso- and metathoracic ganglion (compare e.g. reference 28) and are absent as well from the expression pattern of dsx-GAL4.

Shirangi et al. 2013 (not in our reference list, see: <http://dx.doi.org/10.1016/j.celrep.2013.09.039>), describes the fruitless positive hg1 wing motor neuron connecting to the doublesex positive **wing muscle** hg1. There are no doublesex positive wing muscle motor neurons described in Shirangi et al. 2013.

In Shirangi et al. 2016 (reference 9), the doublesex positive **interneuron** TN1 is described. This neuron is male specific and not present in females. Shirangi et al. 2016 does not describe any dsx positive wing motor neurons.

From our data as well as the literature, we conclude that there are no dsx positive wing motor neurons.

Reviewers' Comments:

Reviewer #1:

None

Reviewer #2:

Remarks to the Author:

I'm happy with all the edits and have no further comments.

Reviewer #3:

Remarks to the Author:

The two major concerns, especially the latter one that I mentioned twice, have not been addressed. The authors seems not willing to perform any control experiments for the very crucial experiment that only used one genotype. If only one genotype were used, it should be the wild-type. For example, wingless wild-type females. If they used *dIm>TNT* females, they should also test at least *dIm* control females.

I have seen so many cases that different genotypes even with the same genetic background have behavioral differences. If one makes an essential conclusion based on one genotype, I will worry about its reproducibility.

Changes in the revised version:

Figure 4i and j shows the results of an additional experiment, playback of female song to wingless wild type females remating with wild type males.

Text referring to this experiment in the results, figure legend and in the method section has been added and is highlighted in yellow.

We also added the source data for the graphs shown in Figure 4i,j to the Source data excel file.

Former Figure 4i, now 4k (schematic of model and summary), has been rearranged and small pictograms have been added, without changing content or message.

Reviewer:

The two major concerns, especially the latter one that I mentioned twice, have not been addressed. The authors seems not willing to perform any control experiments for the very crucial experiment that only used one genotype. If only one genotype were used, it should be the wild-type. For example, wingless wild-type females. If they used *dIm>TNT* females, they should also test at least *dIm* control females.

I have seen so many cases that different genotypes even with the same genetic background have behavioral differences. If one makes an essential conclusion based on one genotype, I will worry about its reproducibility.

Reply:

As suggested, we used wingless wild type females to confirm our finding from *dIm>TNT* mute females, which show delayed remating with *Sb/Tm3, Ser* males when receiving playback as compared to silence during their first mating with a wt CS male.

Results from this new experiment are now shown in the new Fig 4i and j. Since wingless wild type females showed earlier high remating with CS wt males, and their remating could not be assessed by genotyping the progeny, we evaluated remating latency by videotaping fly interactions during the first 20 hrs after the first copulation. Playback leads to significantly longer remating latency, consistent with the first experiment.

The effect of female song playback in significantly delaying remating is thus seen in both experiments, despite the difference in overall remating dynamics between the two different mute female/male combinations (*dIm>TNT* females + *Sb, Tm3, Ser* males and wing amputated CS females + CS males). We therefore confident that the reported effect of female song is not an idiosyncrasy of the *dIm>TNT* females + *Sb, Tm3, Ser* males genotype combination.

Reviewers' Comments:

Reviewer #2:

Remarks to the Author:

I agree with Reviewer 3's concern. Genetic background has a huge effect on behaviors and it should always be well controlled. In terms of the playback experiment, since the only variable is the playback sound, the major concern is that the remating effect may be sensitive to the genetic background.

The additional experiment that the authors have done is perhaps not the best experiment that could have been done. The remating effect was measured in a different way. The effect is statically significant, but rather subtle (which is often a concern with behavioral experiment). In addition, the stress factor (from hearing playback) has not been completely ruled out. However, it does directly address the specific concern of the genetic background effect in remating phenotype. I would accept their conclusion, despite with caution.

REVIEWERS' COMMENTS:

Reviewer #2 (Remarks to the Author):

I agree with Reviewer 3's concern. Genetic background has a huge effect on behaviors and it should always be well controlled. In terms of the playback experiment, since the only variable is the playback sound, the major concern is that the remating effect may be sensitive to the genetic background.

The additional experiment that the authors have done is perhaps not the best experiment that could have been done. The remating effect was measured in a different way. The effect is statically significant, but rather subtle (which is often a concern with behavioral experiment). In addition, the stress factor (from hearing playback) has not been completely ruled out. However, it does directly address the specific concern of the genetic background effect in remating phenotype. I would accept their conclusion, despite with caution.

Response to reviewer:

We are glad the reviewer accepts our conclusions- no further comments to add.